# Application of Machine Learning to Assess the Quality of Food Products—Case Study: Coffee Bean

Krzysztof Przybył [1,*], Marzena Gawrysiak-Witulska [1], Paulina Bielska [1], Robert Rusinek [2], Marek Gancarz [2,3], Bohdan Dobrzański, Jr. [4] and Aleksander Siger [5]

1    Department of Dairy and Process Engineering, Faculty of Food Science and Nutrition, Poznań University of Life Sciences, ul. Wojska Polskiego 31/33, 60-624 Poznan, Poland; marzena.gawrysiak-witulska@up.poznan.pl (M.G.-W.); paulina.bielska@up.poznan.pl (P.B.)

2    Institute of Agrophysics, Polish Academy of Sciences, Doświadczalna 4, 20-290 Lublin, Poland; r.rusinek@ipan.lublin.pl (R.R.); m.gancarz@ipan.lublin.pl (M.G.)

3    Faculty of Production and Power Engineering, University of Agriculture in Krakow, Balicka 116B, 30-149 Krakow, Poland

4    Pomology, Nursery and Enology Department, University of Life Sciences in Lublin, Głęboka 28, 20-400 Lublin, Poland; b.dobrzanski@ipan.lublin.pl

5    Department of Food Biochemistry and Analysis, Poznań University of Life Sciences, ul. Mazowiecka 48, 60-623 Poznan, Poland; aleksander.siger@up.poznan.pl

\*    Correspondence: krzysztof.przybyl@up.poznan.pl

**Abstract:** Modern machine learning methods were used to automate and improve the determination of an effective quality index for coffee beans. Machine learning algorithms can effectively recognize various anomalies, among others factors, occurring in a food product. The procedure for preparing the machine learning algorithm depends on the correct preparation and preprocessing of the learning set. The set contained coded information (i.e., selected quality coefficients) based on digital photos (input data) and a specific class of coffee bean (output data). Because of training and data tuning, an adequate convolutional neural network (CNN) was obtained, which was characterized by a high recognition rate of these coffee beans at the level of 0.81 for the test set. Statistical analysis was performed on the color data in the RGB color space model, which made it possible to accurately distinguish three distinct categories of coffee beans. However, using the Lab* color model, it became apparent that distinguishing between the quality categories of under-roasted and properly roasted coffee beans was a major challenge. Nevertheless, the Lab* model successfully distinguished the category of over-roasted coffee beans.

**Keywords:** artificial intelligence; deep learning; convolutional neural networks (CNNs); coffee bean

## 1. Introduction

Arabica coffee is a species of coffee tree with small beans and belongs to the *Rubiaceae* family. This unique species characterized by the features of an evergreen tree [1], which, through proper breeding in the zone of tropical areas, allows it to maintain its green leafy state almost throughout the year. Coffee beans are used in the preparation of coffee beverage, which is the second most consumed beverage in the world [2]. However, among the 100 species of coffee, it is Arabica coffee that has the greatest economic importance [2] due to its flavor, aroma, low caffeine content, cultivation requirements and product price, among other reasons.

Arabica coffee is one of many group of food products that exhibit beneficial effects on the human body [3–5]. This is related to its content of key bioactive compounds, such as caffeine, lipids, polyphenols, vitamins, minerals and aroma compounds [3,4,6]. In the literature, the consumption of Arabica coffee has been shown to help neutralize free radicals in the body, reducing the risk of chronic diseases including cardiovascular disease [7,8], Alzheimer's and Parkinson's disease, and liver disease [9–12].

The coffee bean also represents a valuable dietary source due to, among other things, the chlorogenic acids (CGAs) they contain [13]. This is a group of polyphenols found naturally in coffee beans. They are also a powerful antioxidant, which, as mentioned, in addition to inhibiting certain diseases, can promote weight loss by, among other things, affecting metabolism and glucose absorption [14,15]. It seems crucial to assess the quality of coffee beans due to the impact of sourcing the right beans during the coffee roasting process, where the quantity and quality of the CGA they contain can degrade rapidly. Therefore, the process of sourcing and selecting the right beans is important in analyzing the health properties of coffee. The interest in health properties has attracted the attention of scientists as well as healthy eaters [3,4,11,12,14,16–18].

Artificial intelligence methods have gained great importance in the food production and distribution process through process optimization, crop monitoring and management, quality assessment [19,20] and food prediction, among others [21].

One of the artificial intelligence methods includes deep learning, which is a technique that uses artificial neural network models containing, among other things, two or more layers of hidden neurons. Deep learning models allow for data analysis without the researcher knowing the exact rules. This network topology processes information and is a tool that offers the mapping of the complex relationships occurring between the analyzed data [20,22]. Actions such as extracting the shape, color or texture of an analyzed object in a bitmap image requires a large amount of data, which translates into longer processing time and statistical analysis of that object. The application of deep learning methods makes it possible to automate some of the activities, among others, related to image data processing at various stages of statistical analysis [22–24].

The convolutional networks, also called convolutional neural networks (CNNs), are a case of deep learning that analyze individual areas of the input dataset and generate new tensors based on them [25]. By means of convolution operations, CNNs use a kernel function and involve increasing the ordinality of copies of the tensor with which operations are performed without increasing their dimensionality [26–28]. Convolutional networks use an algorithm to classify images due to the presence of repeated, recognizable and similar shapes in the image. The input tensor shape of the network accounts for the sample (pattern), image height, image width, and depth as the number of dimensions of the RGB color space [29–31].

In the literature, from the point of view of the problem posed, many researchers have tried to apply CNNs using digital images to assess the quality of green coffee beans during harvesting, for example, by the team of Huang et al. (2020) [18], among others, who achieved a coffee grading rate of 93%. On the other hand, the team of Pinto et al. (2020) [32], based on green coffee beans, similarly made a quality assessment according to defects, obtaining a grading rate of min. 72.4%. Research was also conducted by Wang et al. (2021) [16] to effectively build an intelligent system based on deep learning and computer vision to assist in the detection of defects, including mold, fermentation, insect bites and crushed coffee beans, obtaining neural models with recognition quality rates of 78% and 93% [16].

In view of a number of research works aimed at improving the evaluation of the quality of harvested coffee beans, the current technological solutions in the process of coffee harvesting and roasting are still being sought to be made more effective.

In this work, as mentioned above, the focus was on recognizing three quality classes of coffee beans because of the coffee roasting process. The assumption of the study was to assess the authenticity based on the observation of the color properties of selected Arabica coffee samples obtained through the processing of digital images. The utilitarian purpose was to develop convolutional networks to assist in the identification of quality classes of Arabica coffee beans. The application of modern methods using the Python 3 environment seems to be a key aspect. This is due to the fact that it is easier to implement this type of solution in the food industry than traditional statistical methods. The designed

convolutional models, according to the machine learning criterion, have high scalability, which makes it possible to adapt to applications in the food industry.

## 2. Materials and Methods

### 2.1. Samples

The research material was Arabica coffee beans, which came from the San Pedro Necta area located in the Huehuetenango region of western Guatemala [33]. Specific information about the research material was included in the materials of the research paper by Rusinek et al. (2022) [4,33]. In order to conduct the experiment, 1 kg of each type of coffee was obtained, i.e., heavily roasted coffee (overdeveloped class), weakly roasted coffee (underdeveloped class) and properly roasted coffee (standard class), which was obtained by roasting coffee beans at the Rovigo Coffee roaster (Lublin, Poland). The beans were roasted in a Coffed SR 5 roaster procedure according to Rusinek et al. (2022) [4].

The coffee beans of each test sample (class) were subjected to an image acquisition process. Series of digital images were taken for the tested classes, i.e., coffee beans. Examples of test samples (research classes) describing each type of coffee bean are shown in Figure 1, where Figure 1a shows the equivalent of a properly roasted coffee bean (standard), Figure 1b defines an over-roasted coffee bean, and Figure 1c shows the pattern of an under-roasted coffee bean.

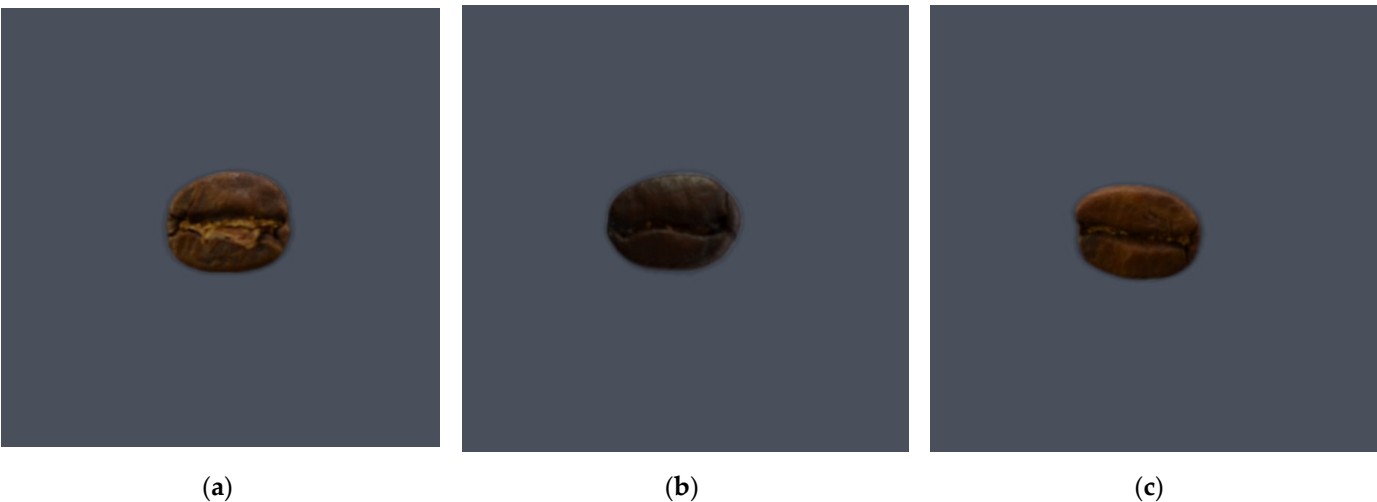

|        |        |        |
|:------:|:------:|:------:|
| (**a**) | (**b**) | (**c**) |

**Figure 1.** Examples of samples of selected coffee beans, considering the quality status: (**a**) standard coffee, (**b**) overdeveloped coffee, (**c**) underdeveloped coffee.

### 2.2. Color Analysis

In this research, a color analysis was performed by determining the L, a, b component parameters of the L*a*b* color space model [34–36] using an X-Rite SP-60 spectrophotometer (X-Rite, Grand Rapids, MI, USA). In order to perform the experiment, coffee beans were placed on plastic Petri dishes. This experiment was performed precisely by carrying out 15 repetitions for each test class of coffee. In analytical methods as well as machine learning, it is important to obtain a reliable number of repetitions to obtain reliable results.

The second step involved performing color analysis with the RGB (Red Green Blue) model, which is more popular in machine learning due to its digital solutions [37,38]. Data on the R, G, B components for the RGB model were extracted from digital images of each coffee research class. In order to extract the necessary information about the RGB color space model contained in the image, the available Python environment supported by the open source libraries Pandas, numPy and Python Image Library (PIL) [39] was used. The PIL library enabled processing and at the same time influenced image modification by acquiring the necessary image pattern, i.e., the extracted object in the form of coffee beans. In order to extract numerical data from the image, the mentioned numPy library was used

to determine measures and statistics of the RGB model. The Pandas library allowed for the analysis of the aforementioned data extracted from the image showing them in a clear (readable, tabular order) manner, which significantly proper representation of numerical data emerges as a key aspect in data preprocessing.

### 2.3. Image Collection by Camera

In order to acquire digital images, a NIKON D5100 (Nikon Corp., Bangkok, Thailand) camera was used, which is characterized by high performance due to its resolution, the speed of capturing images and precise focusing. This camera was equipped with a 16.2-megapixel sensor, which fully ensured the acquisition of sufficient detail when taking digital photos with a coffee bean subject. In addition, this camera takes digital photos at a speed of about 4 frames/s, which also meets the expected criterion for using this model in the study. This camera, the NIKON D5100, was equipped with an AF system that allows for precise focusing. The application of vision devices in various aspects of research requires key image parameters, such as proper focus setting, matching ISO sensitivity value, exposure time selection, and maintaining conditions of the same environment within the repeatability of results. It is very common to encounter the problem of incorrect preparation of image acquisition parameters, which nowadays for machine learning can be a key aspect in the process of image recognition.

Image acquisition using the research and measurement station in the study was performed according to the procedure of the author's solution (see Przybył K. et al. (2023)) [40–42]. In the process of image acquisition, the NIKON D5100 camera was equipped with a Nikkor Lens 28 mm. The selected lens is a wide-angle lenses, which gives the camera to make a wider field of view without losing the quality of the acquired image. The main criterion for the selection of the lens was to be able to take digital images at close range while preserving the quality of the coffee bean structure as well as its proportions. A series of 160 digital images depicting the selected type of coffee were taken during the study. Each of the 160 images contained a single object representing a coffee bean. Each coffee bean, in the process of image acquisition, needed to be rotated to image the top as well as the bottom part. These parts of the coffee bean are significantly different. This resulted in the preparation of 80 coffee beans for each research class.

With the purpose of maintaining the reproducibility of the results, the same image acquisition conditions were established for the coffee beans according to the exposure criterion and the established environment. Before taking digital images, the device was calibrated, and the image parameters were set, including setting the ISO sensitivity at 200, the aperture was set at f/6.3, the exposure time was 160 s, and the focal length was 28 mm. During the process of image acquisition, flash was not used due to the preservation of the conditions regarding the uniformity of illumination established by the measuring and research station according to Przybył K. et al. (2023) [40]. When taking the images, the white balance was set in manual mode in order to achieve the most accurate match of color to the designated environment.

As a result of image acquisition, 240 coffee beans were randomly sampled, i.e., 80 under-roasted coffee beans, 80 over-roasted coffee beans and 80 standard coffee beans (of appropriate quality). In effect, 480 digital images with a resolution of 4928 × 3264 (300 dpi) and 24-bit image depth saved in .JPG format were obtained.

### 2.4. Preprocessing Data

This step, in machine and deep learning, required preparing the data accordingly. Acquired digital photos with a resolution of 4928 × 3264 required cutting the background from the image. Using the Python environment, a catalog of unprocessed (original) photos was loaded in the coffee bean contained in each image. The rembg library in the Python environment was used. Nowadays, machine learning with Python using the rembg library allows for more precise background removal than previously used methods known to the author Przybył K., e.g., in C# programming. The rembg library is open source and supports

various image file formats. In the research activity, 480 digital images in .JPG format were subjected to the process of cutting out the background and saving the files to .PNG format. The transformation of the image to .PNG file format allows for the lossless compression of the data without losing information during compression. Finally, .PNG files also support a transparency channel, allowing images with transparent areas (such as backgrounds) to be extracted, and are supported on multiple platforms.

In the next step, an image cropping process was carried out to create a square image with a lower resolution, i.e., 678 × 678 (secondary image). The reason for changing the image resolution in machine learning became limited computing power for the CNN learning process. Two sets of secondary images were prepared and divided into a learning set and a test set. The testing set defined 3 classes, i.e., 3 types of coffee beans (over-roasted, under-roasted and standard). In total, the testing set contained 480 digital images. In accordance with the principle of machine learning in the learning set, it was required to obtain images that would differ from the data contained in the testing set [43–45]. For this purpose, a point transformation of the image was carried out with steps of 90, 180 and 270 degrees. The result was the acquisition of 1440 digital images for the learning set. Remembering the criterion of over-fitting the model, the following ratio of sets was adopted: 75:25.

### 2.5. Building Neural Model by Python

This step required the design of a CNN model. A diagram of the model development is shown in Figure 2. A CNN was prepared in the Python environment, consisting of input layers, such as convolution layers (Conv2D), max pooling layers (MaxPooling) and a global averaging layer (GlobalaAveragePolling2D), and an output layer (Dense) with 3 neurons, representing the selected class of coffee beans. The Conv2D layers, as the base layers in the CNN, determined the patterns using filters to move the window (convolution window) by a defined step (stride) across the coffee bean image. The MaxPooling layers significantly extracted the most important representative features from the image, affecting the change in the dimension of the objects as well as the computational complexity. The GlobalAveragePolling2D layer was used to reduce the dimensionality of the data contained in the image while still maintaining relevant information. A digital image in the RGB color space model was reduced by calculating the average of all pixel values on a given channel. The reduction in the image information on each channel of the RGB color model, compared to fully connected, prevented, among other things, the overfitting of the model (overfitting) and so-called gradient fading, resulting in further deepening of the CNN for better performance. Dense layer (Dense) usually used for multiclassification, i.e., for a dataset such as coffee beans, where there are a minimum of 3 categorical classes, in this case, overdeveloped, underdeveloped and standard. The output neuron of the output layer is responsible for each class representing the mentioned type of coffee. In the classification issue, the Dense layer transforms the information compacted in the image into a set to predict their membership in certain classes.

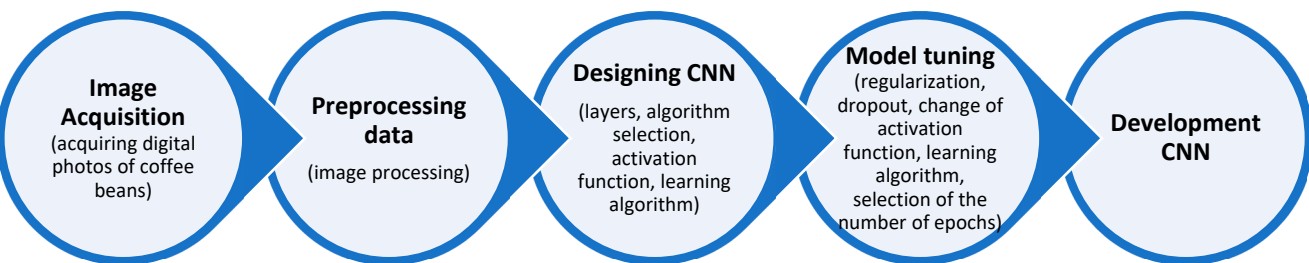

**Figure 2.** A procedure scheme for preparing convolutional neural networks.

*2.6. Statistical Analysis*

In the study, the homogeneity of coffee beans by research class was carried out using analysis of variance (ANOVA), and Tukey's test [46] was applied to more accurately depict mean values between coffee classes using Statistica version 13.3 (Statsoft, Tulsa, OK, USA). Averages for each coffee class were calculated. The critical value of the Tukey test was set at a significance level of $\alpha = 0.05$. The ranges of differences for each class of coffee bean were calculated and compared. As a result, statistical analysis made it possible to accurately assess significant differences between the coffee classes studied. This translated into the extraction of significant characteristics of each coffee class and the establishment of detailed homogeneity for these classes.

## 3. Results and Discussion

*3.1. Color Analysis*

In Tables 1 and 2, the results of the L*a*b* and RGB color space models for each class of coffee are summarized. As a result of analysis by the Tukey test, the existence of subclasses for coffee beans was confirmed. Indeed, the test classes differ from each other [34]. In Table 1 on L*a*b* color, it was observed that statistically, the greatest difference in color was shown by the burnt coffee class against the other classes. Analyzing the L*, a* and b* components of the L*a*b* model [47,48], the under-roasted and standard classes showed a statistically significant similarity between them. In fact, these classes have similar values in the color attributes, which translates into difficulty in distinguishing them from each other. In the case of the over-roasted coffee class, even organoleptic differences in brightness (L*), red–green (a*) and yellow–blue (b*) can be effectively discerned [34].

**Table 1.** Mean of values of L*a*b* color of coffee bean.

| Class of Coffee | L* | a* | b* |
|---|---|---|---|
| underdeveloped | $18.07 \pm 2.03$ [b] | $7.86 \pm 0.51$ [b] | $10.83 \pm 0.84$ [b] |
| overdeveloped | $15.38 \pm 2.31$ [a] | $5.86 \pm 0.58$ [a] | $6.44 \pm 1.16$ [a] |
| standard | $20.13 \pm 3.69$ [b] | $8.13 \pm 0.75$ [b] | $11.81 \pm 1.61$ [b] |

a,b: The differences between mean values with the same letter in columns were statistically insignificant ($p < 0.05$).

**Table 2.** Mean of values of RGB color of coffee bean.

| Class of Coffee | R | G | B |
|---|---|---|---|
| underdeveloped | $0.33 \pm 0.06$ [a] | $0.24 \pm 0.05$ [a] | $0.16 \pm 0.03$ [c] |
| overdeveloped | $0.42 \pm 0.09$ [c] | $0.27 \pm 0.05$ [c] | $0.18 \pm 0.04$ [b] |
| standard | $0.46 \pm 0.08$ [b] | $0.30 \pm 0.05$ [b] | $0.19 \pm 0.03$ [a] |

a–c: The differences between mean values with the same letter in columns were statistically insignificant ($p < 0.05$).

On the other hand, Table 2, in which the results of the RGB color space model are presented [37], shows that the under-roasted coffee class shows a statistical significance in the greatest differences between the other classes. In the case of this model, there are clearly statistically three different groups that show differences between each other [34]. The knowledge that the RGB model is based on the light reflected in a fixed environment may prompt an appropriate representation of these colors for a food product in reality. In having an accurate representation of these colors using the RGB model for a food product, there is greater quality control and it leads to maintaining the authenticity of that product. Currently, techniques are being used to adulterate food products due to their color to meet consumer expectations and at the same time extend their shelf life [49,50]. The RGB color space model, together with machine learning, may soon become a key aspect for determining the quality grade and checking the authenticity of food products. As a result of image acquisition, the dependence on illumination has been established, and this allows for the adjustment of the visual presentation with the help of the RGB model, minimizing the differences between the digital image and the actual visual impression. Nowadays, through

increasing consumer awareness, it is also required that a food product contain information as described for authenticity in the production and the distribution of products [21,51]. By the same means, the RGB color space model, by distinguishing more precisely between different classes of coffee beans than the L*a*b* model, can be of greater benefit in assessing the quality of these products.

*3.2. The Design and Learning Process of Convolutional Networks*

While designing the CNN model, the architecture of the network was determined, with which it was possible to perform the intended task, i.e., the classification of coffee beans, taking into account the mentioned categorical classes. The CNN used a sequential model (Sequential model), with which the number of individual layers was determined. The advantage of the chosen model is its linear flow between layers in convolutional networks [52,53]. Similarly, to the Multilayer Perceptron model, it is the most common and relatively simple type of neural model in machine learning [52], in which there is no feedback, and there is a relatively simple action for adding layers in the CNN structure.

The designed Sequential model contained eight input layers, including four convolution and four MaxPooling layers, one global average layer and one output layer. The convolution window in the convolutional layers was shifted by two units instead of one, which resulted in a faster dimensionality reduction in large tensors in the network layers. After a series of convolution and reduction operations, the feature tensor was transformed to flat link to the dense layer for the final classifier [29,54]. For the output layer, a sigmoid function was used, as it gave better results than the more commonly used softmax function in convolutional networks [39]. For the input layers, the Rectified Linear Unit (ReLU) function was used, which has emerged as the more effective activation function in this issue as in other studies [55]. This is due to the fact that the ReLU algorithm relative to other functions, including sigmoidal and hyperbolic tangent (tanh), prevents gradient from fading when learning convolutional models as well as other types of neural networks.

In the problem of minimizing the loss function during model learning, the Adam algorithm was used with a loss minimization of 0.001. The Adam learning algorithm as well as others [56,57], including RMSprop [58,59], Adadelta and Nadam [60,61], are based on performing network parameter updates [62]. The process of updating the network parameters is based on scaling by the square roots of the squares of the moving averages of the previously quoted gradient values at the point under study. These algorithms are constantly being improved by, among other things, giving them "long-term memory" against the gradient values that characterized the trained model in previous learning epochs [63].

The final step required determining model fit using epoch selection. The most efficient learning of the CNN model finally determined 55 epochs. The results of the learning are shown in Table 3.

**Table 3.** CNN's learning results.

| Type | Testing Set | | | Training Set | | | Activation Hidden | Training Algorithm |
|------|------|------|------|------|------|------|------|------|
| | MSE | MAE | $R^2$ | MSE | MAE | $R^2$ | | |
| CNN | 0.1739 | 0.3238 | 0.8125 | 0.1941 | 0.3344 | 0.7312 | ReLU | Adam |

In the classification process of various research aspects based on bitmap images, the use of convolutional artificial neural networks made it possible to achieve a matching power coefficient in the range above 0.7. A matching power coefficient value below 0.7 would mean that there were discrepancies between the results of the designed convolutional networks and the pattern results for these networks (targets), i.e., they were not fully satisfactory at the specified stage of research. A quality matching coefficient of less than 0.6 would mean that the matching of the convolutional networks is insufficient for the CNNs

to be able to independently decide coffee class recognition with sufficient accuracy. In Table 3, the model obtained was a CNN with a good fitting ability of $R^2 = 0.81$ (Figure 3a). It means that this model satisfied the criterion of strength and the quality of neural network fitting for the indicated research question. In comparison, Mean Square Error (MSE) and Mean Absolute Error (MAE) were determined to assess the quality of the model in machine learning. In Figure 3b, the decreasing value of MSE indicates a more accurate prediction of the CNN model, which translates into optimal model fitting accuracy. The determined MAE value shown in Figure 3c for the training and test sets, respectively, showed greater robustness to outliers than the MSE.

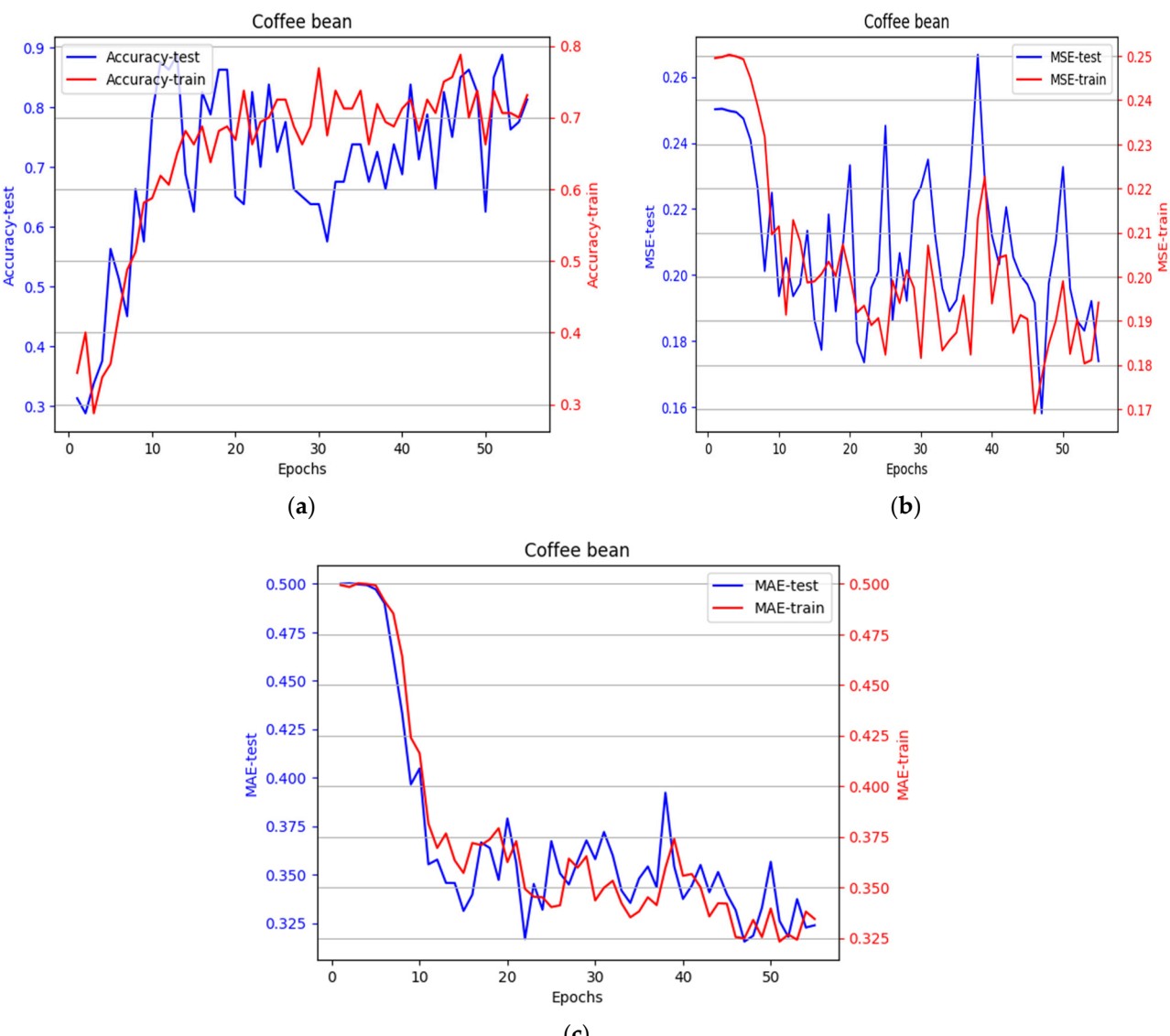

**Figure 3.** Curves for evaluating the quality of the CNN model considering the accuracy index (**a**), MSE (**b**) and MAE (**c**) for the test and training set.

Currently, it was accepted that the usage of deep learning models with multiple hidden layers provides more capabilities than a comparable model composed of applied sequentially added neurons in a single hidden layer in a shallow learning network would. The ability to learn representations and transfer learning to other tasks becomes a key aspect in deep learning, which speeds up the process of learning new sets. In machine learning, the criterion of preventing overfitting is important, which translates into the ability to generalize using methods such as dropout or batch normalization, among others [56,57]. In

order to reduce (overfitting), the designed CNN model was also enriched with a dropout layer. A step of 0.5 was set for the Dropout layer, which translated into improved efficiency to generalize the model, giving a better result on the test set (Figure 3a–c).

In machine learning and deep learning, it was also important to use weight regularization, by means of which it becomes possible to reduce the complexity of the artificial neural network model by taking only low values of modulating variables [64,65]. This is obtained by adding a certain value called cost to the loss function of the network [64]. In this study, an L2 regularization was used for the CNN model, for which the cost value is proportional to the square of the values of the weighting coefficients. The penalty coefficient, for which a value of 0.01 was assumed, allowed for the countering the strength of the regularization of the weights of the learned model, resulting in better performance on the test set for the CNN model (Figure 4).

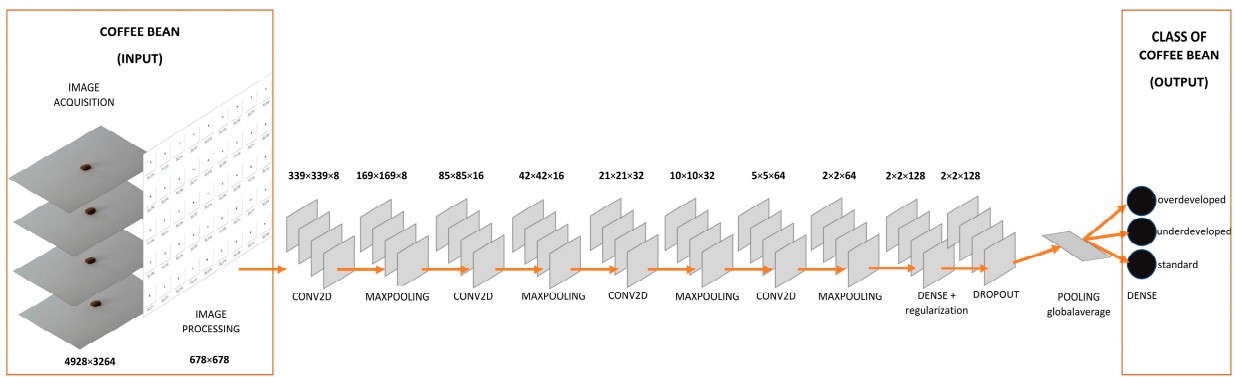

**Figure 4.** CNN model structure includes input data (obtained from image acquisition and processed as a result of image processing) and different types of layers, such as four convolutional layers (CONV2D), four layers of Max-pooling (MAXPOOLING), one dense layer with regularization (DENSE + regularization), one dropout layer (DROPOUT), one global average pooling layer, and describes the output size of the dense layer (DENSE).

## 4. Conclusions

The statistical analysis by ANOVA for color helped distinguish exactly three classes of coffee beans in the RGB color space model, which translated into the efficient recognition of coffee defects obtained during the coffee roasting process. The L*a*b* color model demonstrated the difficulty in distinguishing between the quality classes of under-roasted and properly roasted coffee beans. Nevertheless, the L*a*b* model successfully distinguished between the class of over-roasted coffee. Analyzing the properties of coffee color parameters as an indicator of authenticity can make valuable contributions to food quality and safety.

More challenging was the classification based on digital images with a resolution of 678 × 678 by, among other things, the multidimensionality and complexity of the information contained in the acquired image as a result of image acquisition. The application of deep artificial neural networks made it possible to classify coffee bean samples using computer vision based on image pixels. As a result, an adequate CNN model was acquired, which was characterized by a high fit factor for the test set of 0.81.

The quality assessment of coffee on the basis of representative characteristics (in this case, color) significantly affects both producers and consumers. The effective evaluation of coffee quality will maintain a high product standard in the industrial coffee-roasting process. Consumers will be inclined to select and consume the right kind of coffee through taste and satisfaction with the obtained high-quality food product.

In the future, it is planned to use artificial intelligence methods to help effectively optimize the coffee quality evaluation process. Effectively modeling, operating a series of simulations, making predictions, optimizing, conducting quality control, detecting patterns and diagnosing through machine learning is becoming increasingly important in the food

industry. The use of machine learning algorithms will help in the production of food products, leading to greater speed and efficiency, which will translate into minimizing losses and, at the same time, increasing sales.

**Author Contributions:** Conceptualization, M.G.-W. and K.P.; methodology, K.P. and M.G.-W.; software, K.P.; validation, K.P; formal analysis, P.B., R.R. and B.D.J.; resources, K.P.; data curation, K.P., M.G.-W. and M.G.; writing—original draft preparation, K.P.; writing—review and editing, K.P., M.G.-W., R.R. and A.S.; visualization, K.P.; supervision, M.G.-W. and K.P.; project administration, M.G.-W.; funding acquisition, M.G.-W. and K.P. All authors have read and agreed to the published version of the manuscript.

**Funding:** This study was funded by the Ministry of Education and Science (Poznań, Poland), MEN-UPP 506.784.03.00/KMIP.

**Institutional Review Board Statement:** Not applicable.

**Informed Consent Statement:** Not applicable.

**Data Availability Statement:** The data presented in this study are openly available in Repository for Open Data—RepOD at DOI: https://doi.org/10.18150/0B46MT.

**Conflicts of Interest:** The authors declare no conflict of interest.

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
