# Peer review of "Application of Machine Learning to Assess the Quality of Food Products—Case Study: Coffee Bean"

_applsci, doi:10.3390/app131910786_

Round 1
Reviewer 1 Report
Manuscript is well drafted and results are properly discussed. I recommend this manuscript for publication after incorporating these minor comments.
1. Are the images shown in figure 1 real time pictures?
2. The manuscript is looks great and results are also up to the expectation.
3. One image of software screen should be included performing the testing of the application in real time.
4. One latest article can be cited
https://www.mdpi.com/2227-9717/11/6/1720

Language is scientific and well articulated
Author Response
Dear Reviewer 1,
thank you very much for your valuable comments. I have made the changes as you recommended. This certainly improved and strengthened the substantive aspects of my study.
Response to Reviewer 1 Comments:
Manuscript is well drafted and results are properly discussed. I recommend this manuscript for publication after incorporating these minor comments.
- Are the images shown in figure 1 real time pictures?
The images are not real-time images. As part of the research activities, they were taken for each class immediately after the coffee bean roasting process. However, in the future, it is planned to use the prepared machine learning algorithm to improve the quality assessment of the current coffee roasting system. Moreover, the use of machine learning algorithms will allow to assist in the production of food products much faster and more efficiently, which will translate into minimizing losses and at the same time increasing sales.
- The manuscript is looks great and results are also up to the expectation.
Thank You for the reviewer's accurate opinion.
- One image of software screen should be included performing the testing of the application in real time.
The authors believe that as part of the readability, reliability and at the same time confirmation of the real-time training process, they will make available in the Repod UPP database the files, i.e.: the learning history of the model in .csv file format and the model in .h5 extension format. The data link will be publicly available in the Repod UPP database, which can also be cited. The link has been posted in the Data Availability Statement section: https://doi.org/10.18150/0B46MT.
- One latest article can be cited
https://www.mdpi.com/2227-9717/11/6/1720
We added this item of literature. We corrected into the text.
Kind regards,
Krzysztof Przybył

Reviewer 2 Report
Manuscript entitled "Application of machine learning to assess the quality of food products. Case study: Coffee Bean” is acceptable with minor revisions. Please check comment below
Abstract: Please include brief methods and results in abstract, do not include too much introduction, last sentence should summarize overall results
· Line 32: Remove this sentence “The research subject was Arabica coffee beans”
· Line 47: The “CGAs”, do not start sentence with abbreviation
· Line 56: Remove “In recent times” better write specific years
· Line 61: Cite reference
· Line 234: 3.1. Color analysis: Compare and discuss your results with published literature
· Results and Discussion: Please compare your results with published literature
· Figure 4: Include more explanation for better understanding
· Conclusion: Please summarize conclusion
Author Response
Dear Reviewer 2,
thank you very much for your valuable comments. I have made the changes as you recommended. This certainly improved and strengthened the substantive aspects of my study.
Response to Reviewer 2 Comments:
Manuscript entitled "Application of machine learning to assess the quality of food products. Case study: Coffee Bean” is acceptable with minor revisions. Please check comment below
Abstract: Please include brief methods and results in abstract, do not include too much introduction, last sentence should summarize overall results
We corrected into the text.
- Line 32: Remove this sentence “The research subject was Arabica coffee beans”
We corrected into the text.
- Line 47: The “CGAs”, do not start sentence with abbreviation
We corrected into the text.
- Line 56: Remove “In recent times” better write specific years
We corrected into the text.
- Line 61: Cite reference
We added references. We corrected into the text.
- Line 234: 3.1. Color analysis: Compare and discuss your results with published literature
We added references. We corrected into the text.
- Results and Discussion: Please compare your results with published literature.
We added citations. We corrected into the text.
- Figure 4: Include more explanation for better understanding
The structure of an adequate neural network model is described in section 2.5. Building neural model by Python describing individual layers and 3.2. The design and learning process of convolutional networks, which is illustrated in Figure 4. However, for clarity, a description has been added to the text for Figure 4. We corrected into the text.
- Conclusion: Please summarize conclusion
We corrected into the text.
Kind regards,
Krzysztof Przybył

Reviewer 3 Report
Article With the Title "Application of Machine Learning to Asses the Quality of Food Products. Case Study: Coffee Bean" Has 12 Pages, 3 Tables and 4 Figures. The authors quote 44 sources links as references.
The abstract is very general and the question is whether it meets the requests of MDPI instructions for publishing. From my point of view, the facts on the methodology should be given here and summarized the results and conclusion. However, in this case it is very brief and non-specific.
In Introduction Chapter, the general goal is listed, but the question is, what is the hypothesis of the experiment and whether there is sufficient answers in Conclusion?
Figure 1 is unfortunately not very clear. It is caused by a white background that shines in the eyes. There is no noticeable difference in ligtness (L*) or hue (H).
Figure 1 is unfortunately not very clear. It is caused by a white background that shines in the eyes. There is no noticeable difference in ligtness (L*) or HUE (H). Coffee beans are also rotated otherwise, it would be much more appropriate to indicate more grains in the picture or compare according to the axis.
Chapter 2.2. It features a CIE Color Space. It's about stars. L*a*b* or c*, h (hue). However, on the lines 243 - 244 are the L, a, b without stars. Is it Hunter LAB? L* is lightness, not brightness (CIE 1976).
Chapter 2.3. It is important, but there is only one source (No. 31). This procedure must be taken as recognized, so one reference/source is insufficient.
From my point of view, it is important for the experiment to indicate the origin of the coffee beans and the style and way of roasting. In my view, information about the previous article (reference No. 4) is irrelevant and reduces the possibility for citation and repetition.
Author Response
Dear Reviewer 3,
thank you very much for your valuable comments. I have made the changes as you recommended. This certainly improved and strengthened the substantive aspects of my study.
Response to Reviewer 3 Comments:
Article With the Title "Application of Machine Learning to Asses the Quality of Food Products. Case Study: Coffee Bean" Has 12 Pages, 3 Tables and 4 Figures. The authors quote 44 sources links as references.
The abstract is very general and the question is whether it meets the requests of MDPI instructions for publishing. From my point of view, the facts on the methodology should be given here and summarized the results and conclusion. However, in this case it is very brief and non-specific.
The authors also and with the opinion of the second reviewer improved the abstract. We corrected into the text.
In Introduction Chapter, the general goal is listed, but the question is, what is the hypothesis of the experiment and whether there is sufficient answers in Conclusion?
The authors believe that they have clarified the purpose of the paper, which is discussed in the last paragraph. We have summarized from the point of view of the importance of machine learning in the conclusion as recommended by reviewer 2. We corrected into the text.
Figure 1 is unfortunately not very clear. It is caused by a white background that shines in the eyes. There is no noticeable difference in ligtness (L*) or hue (H).
For clarity, we have corrected Figures 1a, 1b and 1c by changing the background, using axis alignment and cropping for each coffee bean.
Figure 1 is unfortunately not very clear. It is caused by a white background that shines in the eyes. There is no noticeable difference in ligtness (L*) or HUE (H). Coffee beans are also rotated otherwise, it would be much more appropriate to indicate more grains in the picture or compare according to the
For clarity, we have corrected Figures 1a, 1b and 1c by changing the background, using axis alignment and cropping for each coffee bean.
Chapter 2.2. It features a CIE Color Space. It's about stars. L*a*b* or c*, h (hue). However, on the lines 243 - 244 are the L, a, b without stars. Is it Hunter LAB? L* is lightness, not brightness (CIE 1976).
In fact, the authors confirm that this is the L*a*b* color space model. We corrected into the text.
Chapter 2.3. It is important, but there is only one source (No. 31). This procedure must be taken as recognized, so one reference/source is insufficient.
The authors have added more references to literature. We corrected into the text.
From my point of view, it is important for the experiment to indicate the origin of the coffee beans and the style and way of roasting. In my view, information about the previous article (reference No. 4) is irrelevant and reduces the possibility for citation and repetition.
We corrected into the text
Kind regards,
Krzysztof Przybył
